# Novel Homozygous Inactivating Mutation in the *PCSK1* Gene in an Infant with Congenital Malabsorptive Diarrhea

**DOI:** 10.3390/genes12050710

**Published:** 2021-05-10

**Authors:** Laetitia Aerts, Nathalie A. Terry, Nina N. Sainath, Clarivet Torres, Martín G. Martín, Bruno Ramos-Molina, John W. Creemers

**Affiliations:** 1Centre of Human Genetics, Laboratory for Biochemical Neuroendocrinology, 3000 KU Leuven, Belgium; laetitia.aerts@kuleuven.be; 2Department of Pediatrics, Division of Gastroenterology and Nutrition, Children’s Hospital of Philadelphia, Philadelphia, PA 19104, USA; terryn@email.chop.edu (N.A.T.); sainathn@email.chop.edu (N.N.S.); 3Children’s National Medical Center, Director Intestinal Rehabilitation Program, Washington, DC 20010, USA; ctorres@childrensnational.org; 4Department of Pediatrics, Division of Pediatric Gastroenterology and Nutrition, UCLA David Geffen School of Medicine, Los Angeles, CA 90095, USA; MMartin@mednet.ucla.edu; 5Biomedical Research Institute of Murcia (IMIB-Arrixaca), 30120 Murcia, Spain

**Keywords:** proprotein convertase 1/3, *PCSK1*, congenital malabsorptive diarrhea, endocrinopathy, enteroendocrine cells

## Abstract

Proprotein convertase 1/3 (PC1/3), encoded by the *PCSK1* gene, is expressed in neuronal and (entero)endocrine cell types, where it cleaves and hence activates a number of protein precursors that play a key role in energy homeostasis. Loss-of-function mutations in *PCSK1* cause a recessive complex endocrinopathy characterized by malabsorptive diarrhea and early-onset obesity. Despite the fact that neonatal malabsorptive diarrhea is observed in all patients, it has remained understudied. The aim of this study was to investigate the enteroendocrine pathologies in a male patient with congenital *PCSK1* deficiency carrying the novel homozygous c.1034A>C (p.E345A) mutation. This patient developed malabsorptive diarrhea and metabolic acidosis within the first week of life, but rapid weight gain was observed after total parenteral nutrition, and he displayed high proinsulin levels and low adrenocorticotropin. In vitro analysis showed that the p.E345A mutation in PC1/3 resulted in a (near) normal autocatalytic proPC1/3 processing and only partially impaired PC1/3 secretion, but the processing of a substrate in trans was completely blocked. Immunohistochemical staining did not reveal changes in the proGIP/GIP and proglucagon/GLP-1 ratio in colonic tissue. Hence, we report a novel *PCSK1* deficient patient who, despite neonatal malabsorptive diarrhea, showed a normal morphology in the small intestine.

## 1. Introduction

The *PCSK1* gene, which encodes the subtilisin-like proprotein convertase (PC) family member PC1/3, consists of 14 exons and is located on chromosome 5q15–21 in humans [1]. The PC family consists of seven closely related members (furin, PC1/3, PC2, PC4, PACE4, PC5/6, and PC7) and two less-related enzymes (PCSK8 and PCSK9) [2,3]. Because of the high conservation of the catalytic domains of PCs, they all cleave on the C-terminal side of similar but not identical basic amino acid motifs. Despite differences in pH optima, subcellular localization, and tissue distribution of the enzymes, it has remained largely unknown which substrate is cleaved by each PC family member [4]. In addition, some substrates can be cleaved by a specific convertase, while for other substrates, redundancy occurs. Furthermore, this substrate redundancy can even be tissue-specific. For instance, Roebroek et al. reported that the furin knock-out mouse model displayed complete redundancy for the processing of the insulin receptor in liver [5]. In contrast, furin deficiency resulted in the complete blocking of insulin receptor processing in colorectal tumor cells [6].

PC1/3 is expressed at high levels in the hypothalamus, the endocrine pancreas, and the gastrointestinal tract, where it cleaves and activates a number of prohormones and proneuropeptides such as proopiomelanocortin, proinsulin and proglucagon [4]. Due to its processing of intestinal prohormones, PC1/3 plays an important role in the proper functioning of the entereoendocrine system. The entereoendocrine system is the largest endocrine organ in the body and consists of six different cell types: entereocytes, paneth cells, goblet cells, tuft cells, microfold (M) cells, and entereoendocrine cells (EECs). EECs have an open-type morphology and a typical conical shape with the small pointed pole being equipped with microvilli facing in the direction of the gut lumen. On the other side of the cell, the broader base is where secretory granules are situated and release peptide hormones important for appetite regulation and glucose control, such as glucagon-like peptides (GLP-1 and GLP-2), glucose-dependent insulinotropic polypeptide (GIP), and cholecystokinin (CCK) [7,8]. PC1/3 and PC2 are differentially expressed in these EECs, and are responsible for the processing of multiple peptide hormones that can impact gut physiology, such as somatostatin (D cells), CCK (I cells), GIP (K cells), and GLP-1/GLP-2 (L cells) [4,6,9]. ProGIP, which is expressed in the K cells of the small intestine, is activated after cleavage by PC1/3 into GIP. GIP is also called gastric inhibitory polypeptide, referring to its activity as a mild inhibitor of gastric acid secretion [10,11]. Proglucagon, expressed in the α-cells in the pancreas, is processed by PC2, resulting in glucagon. Glucagon is important for increasing blood glucose by stimulation of gluconeogenesis. Proglucagon is also produced in the L cells in the small intestine, where it is cleaved by PC1/3 into GLP-1, GPL-2, and oxyntomodulin [4]. These incretins (GLP-1 and GIP) stimulate a decrease in blood glucose levels by activating insulin secretion by the pancreas, and stimulate satiety by regulating the food-intake regulatory systems in the hypothalamus. GLP-1 and GIP are responsible for 50% of normal insulin release [11]. GLP-1 is also able to inhibit gastric emptying and reduce appetite. GLP-2, which is secreted simultaneously with GLP-1, attenuates gastric emptying and is responsible for villi growth, crypt cell proliferation, and nutrient absorption [12].

Loss-of-function mutations in the *PCSK1* gene cause an autosomal recessive disorder mainly characterized by malabsorptive diarrhea at early stage and childhood obesity when malabsorption is surpassed. In addition, *PCSK1*-deficient patients could have systemic endocrinopathies with incomplete penetrance, such as growth hormone deficiency, hypocortisolemia, and hypothyroidism [4]. The publication of the first *PCSK1* null patient received much attention because of the extreme childhood obesity (36 kg, 3 years old) [13]. Since this index patient and two subsequent patients had been identified on the basis of childhood obesity, this aspect was considered the hallmark of this syndrome, and relatively little attention was paid to the gastrointestinal (GI) phenotype [14,15,16]. This changed when 13 additional null patients were identified in a cohort of only 35 patients with idiopathic malabsorptive diarrhea [13]. The gastrointestinal complications occur immediately after birth, resulting in chronic diarrhea, dehydration, weight loss, and metabolic acidosis, which can lead to death in early childhood. The neonates need prolonged parenteral nutrition and a long hospitalization time. While the malabsorptive diarrhea appears to be lifelong, the parenteral nutrition can be discontinued by 2 years of age. Severe polyphagia reminiscent of leptin disorders is generally observed by late infancy when obesity becomes apparent [13]. To date, 30 cases of *PCSK1* deficiency have been reported, with patients exhibiting a variable range of symptoms (reviewed in [4]). All these patients were diagnosed with early and severe malabsorptive diarrhea and with a consistently elevated proinsulin levels (x8-154 the normal range of 74) [16,17,18,19,20]. There was consanguinity in 81% of the reported cases of *PCSK1*. Four patients died of catheter-related bloodstream infection before the age of 2 years. Characteristic features of PC1/3 deficiency among the diagnosed patients were variable and included: diabetes insipidus (67% of cases), polyuria-polydipsia (85% of cases), postprandial hypoglycemia (52% of cases), growth hormone deficiency (55% of cases), central hypercortisolism (63% of cases), central hypothyroidism (56% of cases), and hypogonadotropic hypogonadism (45% of cases) [15,16,17].

Despite the fact that all 30 published patients presented with features of an enteric endocrinopathy, forming a distinct category of congenital diarrhea and enteropathy (CODEs) disorders, *PCSK1* deficiency remains understudied [13,18]. Here, we diagnosed and biochemically characterized a novel *PCSK1*-deficient patient and investigated the morphology and presence of PC1/3 substrates in the small intestine.

## 2. Materials and Methods

### 2.1. Subject

Genomic DNA was isolated and extracted from blood by standard procedures. In addition, a Clinical Laboratory Improvement Amendment (CLIA)-approved whole exome sequencing was performed on the proband and parents at Children’s Hospital of Philadelphia and subsequently confirmed by Sanger sequencing. The parents signed an informed consent to participate in this study as approved by the Ethical Committee of the University of California, Los Angeles (UCLA).

### 2.2. Histology in Colonic Tissue

Small and large bowel samples were obtained endoscopically. Histology and immunohistological processing of tissue samples was handled as previously described [19]. Secondary antibodies were biotinylated using an ECTASTAIN ABC kit (Vector Laboratories, Burlingame, CA, USA) and diaminobenzidine tetrahydrochloride as the substrate.

Immunohistochemistry staining of the colon tissue of the patient and control patient was performed with antibodies against GLP-1 (Abcam, Cambridge, UK, ab26278; 1:500), GLP-2 (Santa Cruz Biotechnology, Santa Cruz, CA, USA; sc7781, 1:500), *PCSK1* (Abnova, Taipei, Taiwan; H00005122-M02, 1:250), and GIP (Santa Cruz Biotechnology, Santa Cruz, CA, USA; sc23554, 1:250). Staining for Chromogranin A was performed to evaluate enteric dysendocrinosis [20,21].

### 2.3. Site-Directed Mutagenesis

The expression vector pcDNA3 containing the human PC1/3 sequence with a FLAG epitope-tag (DYKDDDDK) between the propeptide and the catalytic domain has been described previously [22]. The mutation p.E345A was generated using the QuickChange site-directed mutagenesis kit (Stratagene, San Diego, CA, USA) following the manufacturer’s protocol. All constructs were verified by Sanger sequencing.

### 2.4. Cell Culture and Transfection Experiments

The human embryonic kidney cell line HEK293T was cultured in DMEM/F12 (Life Technologies, Carlsbad, CA, USA) supplemented with 10% fetal calf serum (Perbio Science, Erembodegem-Aalst, Belgium) at 37 °C and 5% CO_2_. Cells were transfected using Xtreme gene 9 transfection reagent (Roche, Basel, Switzerland) according to the manufacturer’s instructions.

### 2.5. Western Blotting

Cells were lysed in sample buffer (2% SDS, 10% glycerol, 50 mM Tris-HCl pH 6.8), and the conditioned media were collected for methanol precipitation. After protein separation by SDS-PAGE, Western blotting was conducted using an anti-Flag M2 mouse monoclonal antibody (1:10,000, Sigma-Aldrich, Saint Louis, MO, USA). A chemiluminescent substrate was used for the detection of HRP-coupled secondary antibodies using a LAS4000 detection system (GE Healthcare, Chicago, IL, USA).

### 2.6. PC1/3 Activity Measurement

Catalytic activity of PC1/3 was analyzed in conditioned medium from HEK293T cells transiently transfected with the human PC1/3 constructs as described previously [23]. Briefly, 25 μL conditioned medium was incubated at 37 °C with 0.2 mM of the fluorogenic substrate pyr-Glu-Arg-Thr-Lys-Arg-amino methylcoumarin (AMC substrate, 7-amino-4methylcoumarin, pyr-Glu-Arg-Thr-Lys-Arg-AMC) (Bachem, Bubendorf, Switzerland) in 0.1 M Na-CH_3_COOH (pH 5.5), 0.1% Triton X-100, 5 mM CaCl_2_, 1 µM E-64, 1 µM leupeptin, 1 µM pepstatin, and 10 µM TPCK (all from Sigma-Aldrich). Fluorescence was measured every 10 min for 3 h (FLUOstar Omega; BMG Labtech, Ortenberg, Germany).

## 3. Results

### 3.1. Clinical Phenotype Description

The patient, a male infant born to consanguineous parents, weighing 3600 g at birth, developed metabolic acidosis and diarrhea within the first week of life. There was no family history of diarrheal disorders. Although infectious stool studies were negative, the fecal elastase was persistently low: 27–105 μg/g. These results suggested pancreatic insufficiency, but did not improve with pancreatic supplementation. After an acute episode of hypothermia, bradycardia, and lethargy, the patient was diagnosed with adrenal insufficiency and was started on physiologic steroids. He had a low morning cortisol and low adrenocorticotropic hormone (ACTH), as well as an inadequate response to a corticotropin-releasing hormone (CRH) stimulation test. He had normal levels of insulin-like growth factor 1 (IGF-1), insulin-like growth factor-binding protein 3 (IGF–BP3), prolactin, thyroid-stimulating hormone (TSH), and free thyroxine 4 (T4). Growth hormone stimulation testing and MRI of the pituitary was normal. At the age of 22 months, he presented with significant hypernatremia with low urine osmolality in the setting Gram-negative sepsis. He transiently displayed diabetes insipidus when he became septic before stopping IV fluids. A water-deprivation test showed that he did not have diabetes insipidus. Despite low caloric parenteral nutrition intake (20–30 kcal/kg/day with total caloric intake of 70–90 kcal/kg/day) in combination with diarrhea, his weight began to climb between the ages of 10 and 14 months (Figure 1). His height remained stable at the 5th percentile. Premeal proinsulin level was markedly high at 336 pmol/L (normal range: 3–20 pmol/L). A high proinsulin level, congenital diarrhea, adrenal insufficiency, and other endocrinopathies were consistent with a diagnosis of *PCSK1* deficiency [17].

Based on the results obtained during the latest follow-up, sodium serum levels had increased from 137 to 145 (normal 135–145 mmol/L), urine osmolality had increased from 190 to 759 mOsm/kg (normal > 850 mOsm/kg H_2_O), and serum osmolality had increased from 280 to 310 mOsm/kg (normal range for children: 275–290 mOsm/kg). Therefore, there was no need for desmopressin (DDAVP) treatment, and no further evaluation was warranted at that time. Given the fact that children with this condition rarely develop symptomatic adrenal insufficiency, the hydrocortisone treatment was stopped, and his stress dose was increased to 50 mg/m^2^/day. Repeat evaluation of the free T4 levels revealed a lower level of free T4 of 0.97 ng/dL (normal 0.9–1.67 ng/dL), suggesting he had developed central hypothyroidism; this is now being treated with 62.5 μg of levothyroxine on a daily basis. There was one undescended testis present, but the patient did not have a micropenis. Gonadotropins and testosterone levels will be monitored at puberty to see whether he spontaneously goes through puberty or needs testosterone therapy. His IGF-1 levels and body length were normal, so growth hormone deficiency was not suspected. His growth will be monitored.

### 3.2. Gastrointestinal Phenotype

Histology from upper endoscopy and flexible sigmoidoscopy revealed only mild villous blunting without an inflammatory infiltrate. Electron microscopy also revealed normal duodenal structure. Staining for chromogranin A, CD10, and epithelial cell adhesion molecule (EpCAM) was also normal [24]. Stool osmolality was persistently high and fecal-reducing substances were present; thus, malabsorptive CODE disorders were more thoroughly investigated. Breath testing demonstrated malabsorption of sucrose, lactose, and polycose, but normal glucose and fructose absorption (Table 1). Disaccharidase testing was performed on three different occasions and was normal most of the time (Table 2). Diarrhea persisted on milk protein and soy-protein-based formulas, elemental formulas, low-fat, and even low-carbohydrate formula despite the carbohydrate malabsorption. Ultimately, he required total parenteral nutrition (TPN) to maintain electrolyte stability and support his growth. A custom formula with amino acid powder, microlipids, vitamins, electrolytes, and oils were developed, and he was advanced on a hypoallergenic formula. At the age of 3 years, he was able to be weaned from parenteral nutrition.

### 3.3. Functional Analysis of the PCSK1 E345A Mutation

Whole exome sequencing revealed a homozygous c.1034A>C (p.E345A) mutation in exon 8. Multialignment sequence analysis showed that E345, which is located in the catalytic domain of PC1/3, was highly conserved not only between different species, but also between other PC family members (Figure 2A). Visual representation of this residue in the 3D structure of PC1/3 revealed that it was located close to other catalytically relevant amino acids, such as D320 and N309 [25], and to the substrate-binding region, suggesting that E345 could be essential for PC1/3 enzymatic activity (Figure 2A). To validate this hypothesis, we carried out in vitro studies in HEK293T cells transfected with wild-type and mutant human PC1/3. As shown in Figure 2B, PC1/3-E345A was normally synthesized and secreted, without intracellular accumulation of proPC1/3-E345A, suggesting (near) normal autocatalytic activation. On the other hand, carboxyterminal processing of PC1/3-E345A was severely impaired. Although both processes are autocatalytic, propeptide cleavage is an intramolecular process and carboxyterminal processing is intermolecular, which follow different kinetics [26,27,28,29]. Consistently, we did not detect cleavage activity when using a substrate in trans, confirming that this mutant form was catalytically inert (Figure 2C).

### 3.4. Immunohistochemical Analysis of PC1/3 and Potential Substrates in Colonic Biopsies

Tissue staining was performed on the colonic tissue for PC1/3, which was present in a similar cell distribution but qualitatively decreased in the enteroendocrine cells from the patient tissue (Figure 3). PC1/3 has previously been shown to be responsible for the processing of intestinal prohormones such as proGIP [30] and proglucagon [31] in in vitro studies. As shown in Figure 3, immunostaining of GIP and GLP-1 (which are produced by PC1/3 from proGIP and proglucagon, respectively) was similar in control and patient tissue, while GLP-2 (also produced by PC1/3-mediated cleavage of proglucagon in EEC) was more abundant in the patient tissue. Our results indicated that the lack of PC1/3 activity in the intestine did not result in a significative reduction of gut incretins, suggesting that deficiency of other gut hormones was responsible for the malabsorptive phenotype.

## 4. Discussion

In this study, we described a case report of a male newborn with consanguineous parents who presented with congenital malabsorptive diarrhea starting in his first week of life. This hallmark, together with the additional endocrinopathies like hypoadrenalism, reactive postprandial hypoglycemia, diabetes insipidus, and the high proinsulin level, were consistent with a diagnosis of *PCSK1* deficiency. Whole exome sequencing confirmed the diagnosis due to the presence of a novel homozygous mutation c.1034A>C (p.E345A) in exon 8 in the *PCSK1* gene, reported in ClinVar (rs8664309557). A total of 50% of the reported *PCSK1* deficiency cases represented the combination of malabsorptive diarrhea, diabetes insipidus, hypoglycemia, hypercortisolism, and adrenal insufficiency. Early childhood obesity occurred in 79% of the reported cases. Because of the early diagnosis of this new patient, childhood obesity has not yet been observed. As also seen in our novel patient, malabsorptive diarrhea resulted in a need for parenteral nutrition in 92% of cases [32].

The hallmark of *PCSK1* deficiency is the early onset of severe generalized malabsorptive diarrhea. In 100% of the reported cases (26 cases + novel patient), this life-threatening malabsorptive diarrhea is present in the first week/month of life. Martin et al. [13] found 13 pathogenic variants of *PCSK1* after screening a cohort of 35 individuals who suffered from congenital chronic malabsorptive diarrhea. The intestinal biopsy of these cases were normal, with the exception of two patients that exhibited a mild villous atrophy without inflammation [13]. In this new reported patient, despite the appearance of malabsorptive diarrhea in his first week of life, the intestinal biopsy was near normal, with only mild villous blunting without inflammatory infiltrate.

The malabsorptive diarrhea failed to resolve upon selective elimination of fats, carbohydrates, and amino acids; this is the defining characteristic of all enteric endocrinopathies, including *NEUROG3* and *PCSK1* deficiency [13,17,33]. Compared to other published cases of *PCSK1* deficiency, this proband underwent extensive and repetitive assessment of carbohydrate assimilation with breath hydrogen and mucosal disaccharidase testing (Table 1 and Table 2). Significant changes in breath hydrogen levels were seen with disaccharides and polysaccharides but not with monosaccharides; however, elevated reducing substance was detected with all challenges. Abnormalities of disaccharide and polysaccharide assimilation were unrelated to impaired mucosal saccharidase, as the levels were normal in two of the three tests that were performed. Unfortunately, breath hydrogen and mucosal disaccharidase testing is fraught with limitations that diminish their usefulness in assisting in the diagnosis of any of the enteric endocrinopathies. The distinguishing characteristic of all enteric endocrinopathies is the reduced absorptive capacity (as defined by diarrhea) when appropriately challenged with a broad class of nutrients [18,33].

The actual cause of the malabsorptive diarrhea in *PCSK1* deficiency is still unknown, and the available data from the 30 reported patients is limited. Given the well-known function of gut hormones in intestinal physiology, the impaired processing of their precursors is the likely cause. The levels of most gut hormones have not been tested in PC1/3 null patients. The index patient exhibited no elevation of plasma GLP-1 and GLP-2 when tested at the age of 46 (GLP-1: 158 pg/mL (normal: 162 pg/mL)). However, plasma proglucagon seemed dramatically increased (3,430 pg/mL, compared to 328 pg/mL in controls) [16]. Our immunohistochemical analysis revealed that: (1) PC1/3 was still present in the colon of the patient (consistent with the (near) normal synthesis and maturation in the in vitro studies); (2) the expression of GLP-1 and GIP was unaffected by PC1/3 deficiency; and (3) GLP-2 staining appeared to be increased. How proglucagon is processed to GLP-1 in intestinal L cells in the absence of functional PC1/3 is unknown, but furin and PC6A have been shown to be able to perform these cleavages in vitro as well, albeit less efficiently [34]. The ability of PC2 to compensate for the loss of PC1/3 activity is another plausible hypothesis because both PCs are present in the regulated secretory pathway. However, in the α-cells in the pancreas, PC2 activity results in glucagon as the cleaving product, which has the opposite effect of GLP-1 [4]. In order to know whether PC2 is able to produce incretins from proglucagon in the absence of PC1/3 activity in enteroendocrine intestinal cells, further studies are warranted.

Our sequence analysis revealed that the E345 residue, which is located in the catalytic domain of PC1/3, is completely conserved in all species ranging from mammals to invertebrates and all PC family members. Its conservation and location in the catalytic domain suggest that the glutamic acid is important for enzymatic activity. Additionally, the visual representation indicated that this residue is closely located to two other catalytically relevant amino acids, D320 and N309. D320 is important for stabilizing the binding of Ca^2+^ at the S1 binding site [35]. N309 is the oxyanion hole transition state-stabilizing amino acid, which stabilizes the enzyme-substrate tetrahedral intermediate. Wilschanski et al. reported PC1/3 deficiency in three siblings with a novel homozygous N309K mutation [25]. This mutation also leads to normal maturation and secretion but no catalytic activity in vitro. Although the E345A mutant does not cleave substrates in trans, this mutation did not exert any effect on the maturation and secretion of the enzyme, suggesting normal autocatalytic activation. The prodomain of proPC1/3 is cleaved in a rapid intramolecular way at the primary site RSKR110 (t^1/2^ < 2 min) and after it enters the more acidic Golgi compartments, at the second cleavage site RRSRR^81^ [26,27,28,29]. In our in vitro assay, we showed normal cleavage and subcellular trafficking. Apparently, the mutation does not affect normal folding and intramolecular propeptide cleavage, which follows zero-order kinetics. However, further cleavage at the carboxyterminus into two truncated active forms, 74 kDa and 66 kDa, were intermolecular events, following first-order kinetics [4]. In our in vitro study, the truncated forms were not present, which was consistent with the lack of processing of a substrate in trans, which is a second-order reaction.

## 5. Conclusions

In conclusion, we found that the homozygous c.1034A>C (p.E345A) point mutation, present in a pediatric patient with congenital malabsorptive diarrhea, resulted in an inactive form of the PC1/3 protein, and that the gastrointestinal phenotype could be explained, at least partially, by the lack of PC1/3 activity in EECs, although not detecting abnormal processing of intestinal prohormones such as proglucagon or proGIP. Our results suggest that PC1/3 is involved in the processing of other intestinal peptides that might be necessary for a proper nutrient absorption. However, further research is warranted in order to find out whether *PCSK1* deficiency may result in EEC dysfunction.

## Figures and Tables

**Figure 1 genes-12-00710-f001:**
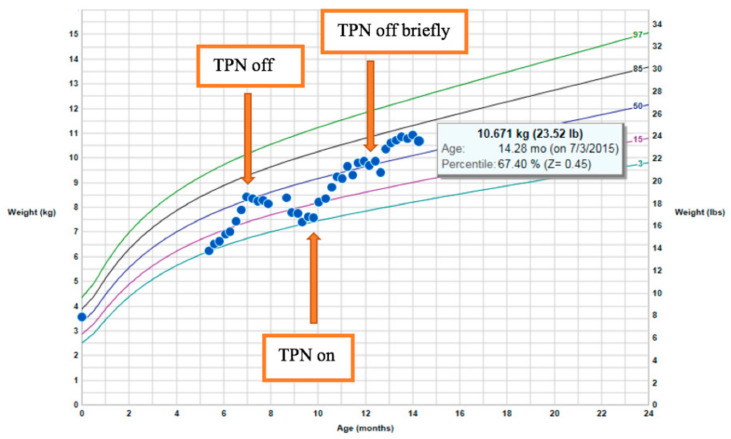
Rapid weight gain on limited total parenteral nutrition (TPN) calories.

**Figure 2 genes-12-00710-f002:**
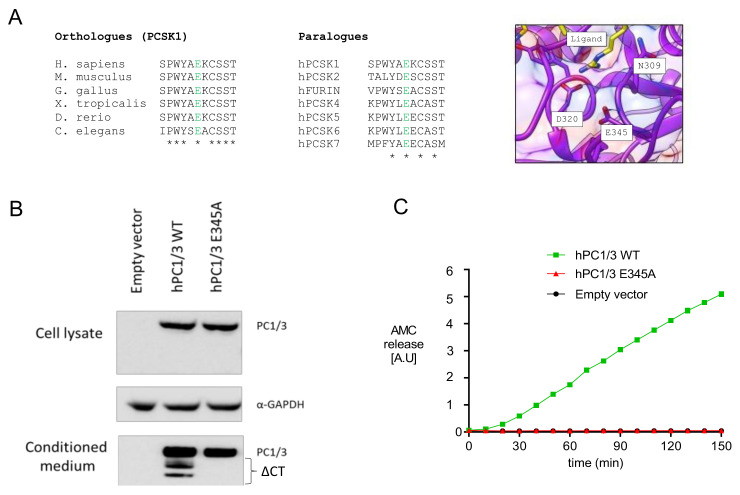
PC1/3-p.E345A has autocatalytic activity but is unable to cleave substrates in trans. (**A**) Alignment of E345 between different species and the other members of the PC family, and visual representation of the of the E345 residue in the 3D structure of PC1/3. (**B**) Maturation and secretion of PC1/3-WT and PC1/3-p.E345A. Western blot of transiently transfected PC1/3-WT and PC1/3-p.E345A in HEK239T cells. GAPDH was used as housekeeping gene. ΔCT: C-terminally truncated PC1/3. (**C**) Enzymatic activity of secreted PC1/3-WT and PC1/3-pE345A was compared using Suc-Leu-Leu-Val-Tyr-AMC as substrate. hPC1/3: human PC1/3; WT: wild-type; AMC: 7-amino-4-methylcoumarin (Suc-Leu-Leu-Val-Tyr-AMC).

**Figure 3 genes-12-00710-f003:**
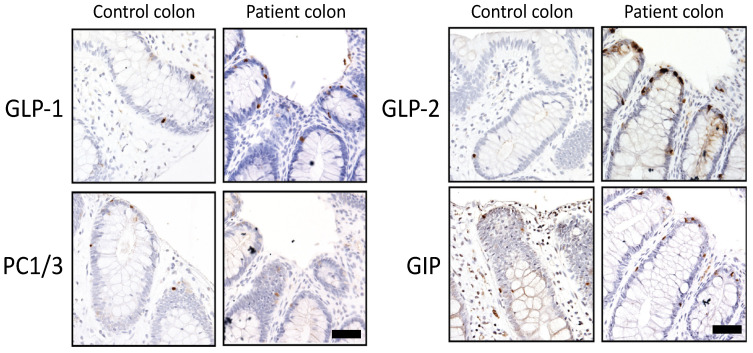
Immunohistochemical analysis of PC1/3 and gut hormones in colonic tissue. GLP-1: glucagon-like peptide 1; GLP-2: glucagon-like peptide 2; PC1/3: proprotein convertase 1/3; GIP: glucose-dependent insulinotropic polypeptide. Scale: 50 μm.

**Table 1 genes-12-00710-t001:** Extensive carbohydrate malabsorption documented by hydrogen breath testing (measured in parts per million).

	Baseline	30 min	60 min	90 min	120 min	150 min	180 min
Glucose	11	8	6	10	17		
Polycose	0	3	3	3	4	28	24
Sucrose	14	23	70	75	107	182	217
Fructose	8	10	13	14	22	21	14
Lactose	12	10	27	55	64	79	34

**Table 2 genes-12-00710-t002:** Mucosal disaccharide analysis.

	μmol/min/g	Test 1	Test 2	Test 3 ^1^
Lactase	14-33	29.3	6.2	18.8
Sucrase	25-66	78.1	26.5	22.4
Maltase	135-205	314.8	112	93.6
Palatinase	8.5-22	23.7	8.9	6.6

^1^ Performed during weeks of breath testing.

## Data Availability

Not applicable.

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
