# Peer review of "Novel Homozygous Inactivating Mutation in the PCSK1 Gene in an Infant with Congenital Malabsorptive Diarrhea"

_genes, 2021, doi:10.3390/genes12050710_

Round 1

Reviewer 1 Report

This is a report describing a novel homozygous inactivating mutation in the PCSK1 gene in an infant. The manuscript is clearly and concisely written, the conclusions are sound and supported by the the presented data.  The authors present convincing functional proof of the mutation effect The  findings are interesting for a broad genetics and developmental biology community. I don't have any major points for revision.Ass minor points I only have text editing/English language as there a handful of grammatical errors in the manuscript.

Author Response

We thank the reviewers for their useful comments. Below we include a point-by-point response to the reviewers’ comments.

Reviewer 1:

Comments and Suggestions for Authors

This is a report describing a novel homozygous inactivating mutation in the PCSK1 gene in an infant. The manuscript is clearly and concisely written, the conclusions are sound and supported by the presented data.  The authors present convincing functional proof of the mutation effect The findings are interesting for a broad genetics and developmental biology community. I don't have any major points for revision. As minor points I only have text editing/English language as there a handful of grammatical errors in the manuscript.

We would like to thank the reviewer for his/her positive comments. As suggested, the manuscript has now been proofread by a professional English editing service. We hope that this new version does not contain any grammatical errors or typos.

Reviewer 2 Report

  1. As the case background of the PCSK1 germline mutation, does the patient have a related family history? And what is the mutation status of patients?
  2. Author should mention that this mutation (rs864309557) has been reported in ClinVar, although the interpretation is of uncertain significance.
  3. Author should reduce the content of PCSK1 biological function in the introduction, which is very well known. 
  4. The authors should show the Sanger sequencing results of the PCSK1 mutation.
  5. In Figure 3, authors should label PC1/3 of the IHC staining, instead of PCSK1, which is the gene name. And scale bar is missed.
  6. Authors should show higher resolution IHC pictures. It is hard to decide PC1/3 is " qualitatively decreased" in the patient's specimen. Also for other protein expression changes.

Author Response

We thank the reviewers for their useful comments. Below we include a point-by-point response to the reviewers’ comments.

Reviewer 2:

Comments and Suggestions for Authors

  1. As the case background of the PCSK1 germline mutation, does the patient have a related family history? And what is the mutation status of patients?

Thank you for the comment. There was no family history of diarrheal disorders. We have included this information in the results section (lines 168-169, page 4).

  1. Author should mention that this mutation (rs864309557) has been reported in ClinVar, although the interpretation is of uncertain significance.

Thanks for the suggestion. We have now included this information in the new version of the manuscript (line 281, page 7).

  1. Author should reduce the content of PCSK1 biological function in the introduction, which is very well known. 

We agree that we could have been more concise and have now shortened the introduction.

  1. The authors should show the Sanger sequencing results of the PCSK1 mutation.

Unfortunately, this image is not available. We’d like to mention that although traditionally Sanger sequencing was needed to substantiate the WES, developments in NextGen sequencing have made this superfluous in many cases. Here, Sanger sequencing was performed as part of the diagnostic routine, even though the WES had sufficient coverage to unequivocally call the mutation without it.

  1. In Figure 3, authors should label PC1/3 of the IHC staining, instead of PCSK1, which is the gene name. And scale bar is missed.

We have now modified Figure 3 as suggested by the reviewer.

  1. Authors should show higher resolution IHC pictures. It is hard to decide PC1/3 is " qualitatively decreased" in the patient's specimen. Also for other protein expression changes.

We have now included higher resolution IHC pictures.

Round 2

Reviewer 2 Report

The authors have clarified my concerns.